



# A new bootstrap technique to quantify uncertainty in estimates of ground surface temperature and ground heat flux histories from geothermal data

Francisco José Cuesta-Valero[1,2], Hugo Beltrami[3], Stephan Gruber[4], Almudena García-García[1,2], and J. Fidel González-Rouco[5]

[1]Department of Remote Sensing, Helmholtz Centre for Environmental Research (UFZ), Permoserstraße 15, Leipzig, 04318, Saxony, Germany.
[2]Remote Sensing Centre for Earth System Research, Leipzig University, 04103, Leipzig, Germany.
[3]Climate & Atmospheric Sciences Institute, St. Francis Xavier University, 5009 Chapel Square, Antigonish, B2G 2W5, NS, Canada.
[4]Department of Geography and Environmental Studies, Carleton University, 1125 Colonel By Dr, Ottawa, K1S 5B6, ON, Canada.
[5]Instituto de Geociencias, Consejo Superior de Investigaciones Científicas - Universidad Complutense de Madrid, Madrid, Spain

**Correspondence:** Francisco José Cuesta-Valero (francisco-jose.cuesta-valero@ufz.de)

**Abstract.** Estimates of the past thermal state of the land surface are crucial to assess the magnitude of current anthropogenic climate change, as well as to assess the ability of Earth System Models to forecast the evolution of the climate near the ground, not included in standard meteorological records. Subsurface temperature data are able to retrieve long-term changes in surface energy balance –from decadal to millennial time scales, thus constituting an important record of the dynamics of
the climate system that contributes low-frequency information to proxy-based paleoclimatic reconstructions. A broadly used technique to retrieve past temperature and heat flux histories from subsurface temperature profiles based on a Singular Value Decomposition (SVD) algorithm was able to take into account a limited number of sources of uncertainty, with recent works attempting to increase the number of factors considered in uncertainty estimates. Nevertheless, the SVD methodology did not define a statistical framework for aggregating inversions of individual profiles to derive global results, which lead to estimates of global and regional uncertainties that are difficult to interpret. To alleviate the lack of a conceptual framework for estimating
uncertainties in past temperature and heat flux histories at regional and global scales, we combine a new bootstrapping sampling strategy with the broadly used SVD algorithm, and assess its performance against the original SVD technique and another technique based on generating perturbed parameter ensembles of inversions. The new bootstrap approach is able to reproduce the prescribed surface temperature series used to derive an artificial profile. Bootstrap results are also in agreement with the global mean surface temperature history and the global mean heat flux history retrieved in previous studies. Furthermore, the
new bootstrap technique provides with a meaningful uncertainty range for the inversion of large sets of subsurface temperature profiles. We suggest the use of this new approach particularly for aggregating results from a number of individual profiles, and to this end, we release the programs used to derive all inversions in this study as a suite of codes labelled CIBOR 1.0: Codes for Inverting BORholes, version 1.0.





## 1 Introduction

Anthropogenic activities have contributed to a sustained increase in the Earth's energy imbalance at the top of the atmosphere (Hansen et al., 2011; Stephens et al., 2012; Johnson et al., 2016; Marti et al., 2022), inducing a radiative response from the climate system (Donohoe et al., 2014). As part of this response, energy exchanges among the ocean, the cryosphere, the continental land masses, and the atmosphere have been altered, leading to an increase in the heat stored in these components of

the Earth System (Hansen et al., 2011; von Schuckmann et al., 2020). The ocean accounts for around 89% of the additional heat storage, with continental landmasses the second largest component storing 6% of the total heat, followed by the cryosphere (4%) and the atmosphere (1%). Monitoring the evolution of the Earth heat inventory is of critical importance to understand the state of the climate system, its associated climate change and future societal and ecosystem risks. Changes in heat storage within each component affect the dynamics of important phenomena, for example heat uptake by the cryosphere contributes to

sea level rise (Oppenheimer et al., 2019), heat gain by the atmosphere affect the development of extreme precipitation events (Pendergrass and Hartmann, 2014), and heat gain by the continental subsurface can increase the release of greenhouse gases from northern soils (Hicks Pries et al., 2017; McGuire et al., 2018).

Most global estimates of continental heat storage have been inferred from subsurface temperature profiles (STPs) (Beltrami et al., 2002; Beltrami, 2002; von Schuckmann et al., 2020; Cuesta-Valero et al., 2021c), which allow to estimate past ground

surface temperature and ground heat flux changes at decadal to centennial time scales (e.g., Shen et al., 1992; Beltrami et al., 2002; Beltrami, 2002; Hopcroft et al., 2007; Demezhko and Gornostaeva, 2015a; Kukkonen et al., 2020). These long-term estimates of surface temperature and ground heat flux changes have also been used to evaluate the ability of general circulation models (GCMs) to reproduce past changes in the conditions of the shallow continental subsurface, which has increased our knowledge of the Earth System and the confidence in future projections (González-Rouco et al., 2006; González-Rouco

et al., 2009; García-García et al., 2016; Cuesta-Valero et al., 2019; Melo-Aguilar et al., 2020). Furthermore, ground surface temperature and heat flux reconstructions from subsurface temperature data have been essential to inform the development of land surface model components, improving the representation of heat transfer through the continental subsurface in climate simulations (Alexeev et al., 2007; Nicolsky et al., 2007; Stevens et al., 2007, 2008; MacDougall et al., 2010; Cuesta-Valero et al., 2016; Hermoso de Mendoza et al., 2020; Cuesta-Valero et al., 2021b; González-Rouco et al., 2021).

Several techniques exist to retrieve the evolution of ground surface temperature changes from STPs, all yielding similar results (Beltrami and Mareschal, 1992; Shen et al., 1992; Hartmann and Rath, 2005; Hopcroft et al., 2007; Cuesta-Valero et al., 2021c). These techniques solve the inversion problem, that is, estimating the surface temperature that generated the observed profile, with two main strategies to retrieve STP inversions: Bayesian methods (Shen et al., 1992; Woodbury and Ferguson, 2006; Hopcroft et al., 2007, 2009), and methods based on a Singular Value Decomposition (SVD) algorithm (Beltrami et al.,

1992; Beltrami and Mareschal, 1992; Clauser and Mareschal, 1995; Hartmann and Rath, 2005; Jaume-Santero et al., 2016; Cuesta-Valero et al., 2021c). Nevertheless, several sources of uncertainty arise in the inversion process, being the most important the unknown thermal properties at most sites, the determination of the quasi-equilibrium temperature profile at each site, and the value of several parameters in the inversion framework, such as the number and length of the time steps for mod-





elling the retrieved surface temperature series, and the number of eigenvalues retained in the SVD algorithm to obtain stable

solutions. Bayesian frameworks allow to easily include different sources of uncertainty in the inversions, while the methods based on applying a SVD algorithm are less flexible, requiring the development of additional sampling strategies to include the different uncertainties in the inversion.

A broadly used method to include the uncertainty due to the unknown quasi-equilibrium temperature profile at each site in SVD inversions consists in performing a linear regression analysis of the deepest part of the observed profile, then providing

with a best estimate of the quasi-equilibrium profile and using the error in the regression coefficients to generate two extremal profiles that constitute the upper and lower limit of the uncertainty range (e.g., Beltrami et al., 2015a, b). The anomaly profiles obtained by subtracting these three profiles to the measured STP are then inverted, providing with a best estimate of the past ground surface temperature history and an uncertainty range given by the inversion of the two extremal profiles. Repeating the SVD inversion using different time step lengths to characterize the surface temperature series and retaining a different number

of eigenvalues in the solution allow to assess the effect of each parameter in the retrieved inversion (e.g., González-Rouco et al., 2009). Nevertheless, including the uncertainty due to unknown thermal properties in the ground into SVD inversions required the development a different approach. This new approach was labelled Perturbed Parameter Inversion (PPI), because it was based on estimating a large ensemble of SVD inversions by changing the value of the inversion parameters for each iteration, including the value of thermal properties (Cuesta-Valero et al., 2021c). After generating the ensemble, the 2.5th, 50th

and 97.5th percentile of all solutions was computed in order to obtain a best estimate and a 95% confidence interval. Thereby, the PPI approach is a generalization of the SVD algorithm, allowing to include the uncertainty due to all important factors described above in the inversions of individual profiles.

Estimates of ground heat flux histories have been retrieved from STPs using two main methods: inversion of subsurface heat flux profiles, and from ground surface temperature histories. Subsurface heat flux profiles can be estimated directly from

measured STPs using the Fourier equation, and due to the nature of heat diffusion through the ground, these heat flux profiles can be inverted following the same SVD approach used to invert subsurface temperature profiles (Beltrami, 2001; Turcotte and Schubert, 2002; Cuesta-Valero et al., 2021c). Nevertheless, subsurface heat flux profiles are noisier than STPs, thus the uncertainty in the retrieved ground heat flux histories is large. The other approach consists in applying the solution of the heat diffusion equation obtained in Wang and Bras (1999) to the ground surface temperature history estimated by inverting the

corresponding STP, reducing the noise in the retrieved ground heat flux history (Beltrami et al., 2002).

Although both SVD and PPI approaches are able to retrieve ground surface temperature and ground heat flux histories from individual STPs, these techniques do not include a meaningful, interpretable framework for aggregating inversions from sets of different profiles. An approach used in the past to obtain regional and global estimates consisted in simultaneously inverting all logs in a given dataset, retrieving an unique ground surface temperature history representing the zonal average

(Beltrami, 2002). Nevertheless, this approach is better used with profiles measured at similar years, because it considers only one surface temperature series, and thus it does not include the effect of the different logging year of each STP on the solution. Alternatively, regional and global averages can be estimated as the mean of ground surface temperature histories (Cuesta-Valero et al., 2021c). However, the uncertainty range for the SVD case is estimated as the mean of the solutions from the extremal





anomaly profiles retrieved for each measured STP, and as the mean of the 2.5th and 97.5th percentiles from each profile in
the case of PPI solutions. These aggregation strategies are not based on sound statistical principles, which makes difficult to
assess the realism of the retrieved uncertainty ranges for regional and global ground surface temperature histories. The same
limitations arise in estimates of regional and global ground heat flux histories obtained with the SVD and PPI techniques.

Here, we introduce a new statistical framework to improve estimates of uncertainty in ground surface temperature and
ground heat flux histories from SVD inversions. This new method is based on a bootstrapping technique to sample the plausible
values of poorly known inversion parameters, together with a broadly used SVD algorithm, that can be applied to any number
of profiles. Results from this new approach, hereinafter bootstrap inversions, are compared against those from the original
SVD method and the PPI approach. An artificial temperature profile and real STP measurements from the Xibalbá dataset
(Cuesta-Valero et al., 2021a) are inverted using these techniques. Using an artificial profile generated from a known boundary
condition allows for evaluating the performance of each method and the choice of parameters against a known surface signal,
before estimating global surface temperature histories and global ground heat flux histories from a large dataset of subsurface
temperature profiles. These experiments also allow to identify shortcomings in the uncertainty ranges derived from the typical
SVD and PPI techniques, particularly when aggregating a large number of profiles. Our results reinforce the role of the STP
inversions as indicators of the long-term evolution of surface conditions.

## 2   Data

### 2.1   Proxy-Based Temperatures

Global temperature reconstructions from those in Figure 1 of the Summary for Policymakers of the sixth Assessment Report of
the Intergovernmental Panel on Climate Change (IPCC) are used here for comparison purposes and to generate synthetic data
to test the inversion methods (Gillett et al., 2021; IPCC, 2021). These reconstructions are based on multi-proxy systems such
as corals and tree rings, and were estimated by the 2k Network of the Past Global Changes organization (PAGES2k) (Neukom
et al., 2019; PAGES2k Consortium, 2022). The global mean temperature anomaly is provided as a smoothed temporal series
using a 10-year window, thus data are available from year 5 to 1995 of the Common Era (CE).

PAGES2k reconstructions combine land and marine proxies to produce global mean temperatures, which we transform into
global land temperatures. To this end, we apply the ratio between land and ocean temperature changes estimated in Harrison
et al. (2015) based on an ensemble of global climate simulations. This multimodel ensemble includes different forcing scenarios
at several time scales, resulting in land surface temperature changes being $\sim 2.36$ times larger than sea surface temperature
changes. Therefore, the PAGES2k global temperature anomaly can be scaled to land temperature changes multiplying by
$\sim 1.38$. The resulting temperature anomaly after the scaling is modified to have 1300-1700 CE as period of reference, the same
as the inversions of subsurface temperature profiles (see below). The land temperature anomaly from the PAGES2k global
temperature anomaly, labelled PAGES2k-Land temperature hereinafter, is used as the upper boundary condition to derive an
artificial subsurface temperature profile designed to test the performance of the different methods considered here. Additionally,
inversions of the Xibalbá database are also compared with these PAGES2k-Land temperatures.





## 2.2 Xibalbá Subsurface Temperature Profiles

Subsurface temperature profiles from the Xibalbá dataset (Cuesta-Valero et al., 2021a, c) are inverted to estimate global mean ground surface temperature histories. Xibalbá profiles were assembled using measurements from several sources, including the

National Oceanic and Atmospheric Administration (NOAA) server (NOAA, 2019) and several publications (Jaume-Santero et al., 2016; Suman et al., 2017; Pickler et al., 2018). Each log was screened to ensure that the profiles do not contain obvious non-climatic signals due to water flow or other factors. All logs were truncated to contain depths between 15 m and 300 m to ensure that all profiles produce inversions relative to the same temporal period (González-Rouco et al., 2009; Beltrami et al., 2011, 2015a; Cuesta-Valero et al., 2019; Melo-Aguilar et al., 2020). The dataset includes 1,079 STPs distributed around the

world, although with a lack of measurements in South America, most of Africa, the Middle East, and southeastern Asia (see Figure 1 in Cuesta-Valero et al., 2021c).

## 3 Methods

The new CIBOR 1.0 (Codes for Inverting BORholes, version 1.0) is a collection of scripts to perform SVD inversions of subsurface temperature profiles, including three different strategies to aggregate results from any number of individual profiles,

which is fundamental to derive regional and global averages of past ground surface temperature and ground heat flux histories. This section explains the general physical and statistical principles to perform SVD inversions of subsurface temperature profiles and to estimate the uncertainties involved in the process, which are included in the CIBOR 1.0 scripts. For information about the accessibility of the codes, please check the Code availability statement at the end of this article.

### 3.1 Subsurface Temperature Profiles

Ground temperatures change with depth as a response to the surface energy balance variations and the heat flow from the Earth's interior, constant at time scales of millions of years (Jaupard and Mareschal, 2010). These variations in the surface energy balance are propagated through the ground following the one dimensional heat diffusion equation, altering the quasi-equilibrium temperature profile resulting from the long-term surface temperature ($T_0$) and geothermal gradient ($\Gamma_0$), that is, the profile that would correspond to consider constant surface conditions. Assuming constant thermal properties through the

ground column, a subsurface temperature profile can be described as (Carslaw and Jaeger, 1959)

$$T\left(z,t\right) = T_0 + \Gamma_0 \cdot z + T_t\left(z\right), \tag{1}$$

where $z$ is depth, the term $T_0 + \Gamma_0 \cdot z$ constitutes the quasi-equilibrium temperature profile, and the term $T_t\left(z\right)$ consists in the signature of recent changes in the surface energy balance.

Equation (1) describes a temperature profile in an homogeneous medium. Although real profiles present changes of thermal

properties with depth, most measurements of temperature profiles provide little or no thermal property data, thus requiring assumptions about thermal diffusivity and conductivity values in order to perform the analyses. Furthermore, subsurface temperature profiles used in climate studies that have adequate measurements of thermal properties have shown that thermal prop-





erties typically vary relatively in a small amount around a mean value with depth, e.g., the Neil Well in the Arctic (Beltrami and Taylor, 1995), which supports the use of constant thermal properties to describe these profiles.

The perturbation of the quasi-equilibrium thermal profile in a homogeneous medium due to a step change ($\Delta T$) in surface temperature at a certain time $t$ in the past can be described as (Carslaw and Jaeger, 1959)

$$T(z) = \Delta T \cdot \mathrm{erfc}\left(\frac{z}{2\sqrt{\kappa \cdot t}}\right),\tag{2}$$

where $\mathrm{erfc}$ is the complementary error function, $\kappa$ is the thermal diffusivity of the medium, and $z$ is depth. Therefore, the final anomaly profile caused by the propagation of a series of step changes ($\Delta T_i$) in surface temperatures through the ground can

be expressed as

$$T_t(z) = \sum_{i=1}^{N} \Delta T_i \cdot \left[\mathrm{erfc}\left(\frac{z}{2\sqrt{\kappa \cdot t_i}}\right) - \mathrm{erfc}\left(\frac{z}{2\sqrt{\kappa \cdot t_{i-1}}}\right)\right],\tag{3}$$

which represents the depth-varying temperature term in Equation (1). $T_t(z)$ consists in the solution of the forward problem, that is, for a given surface temperature time series Equation (3) provides the total perturbation of the subsurface profile in response to changes in the upper boundary condition. Therefore, Equation (3) constitutes a one-dimensional forward model of

heat diffusion through the ground.

### 3.2 Artificial Profile

An artificial STP is generated using the PAGES2k-Land temperature as upper boundary condition for a purely-conductive, homogeneous, half-space forward model (see Equation 3). This forward model generates a temperature profile containing the changes in subsurface temperatures as a response to the prescribed changes in surface temperatures with time from the

PAGES2k-Land temperatures. A quasi-equilibrium temperature profile, consisting in a long-term surface temperature of $8\,^{\circ}\mathrm{C}$ and a long-term geothermal gradient of $20\,^{\circ}\mathrm{C\,km^{-1}}$ is added to the profile, as well as Gaussian noise with zero mean and a standard deviation of $0.02\,^{\circ}\mathrm{C}$ to account for measurement uncertainties in real profiles (Beltrami et al., 2015a). Ground surface temperature histories are estimated from the inversion of the resulting artificial profile (Figure 1) using the three techniques detailed below. These temperature histories are then evaluated against the original PAGES2k-Land temperatures that were used

as boundary conditions to generate the synthetic data to evaluate the performance of each technique.

### 3.3 SVD Inversions

The inversion problem for STPs consists in estimating the magnitude of the past changes in surface temperature that gave rise to the observed variation of temperature with depth. Such changes in surface temperature are typically modelled as a series of step changes whose magnitude is determined by the inversion technique, with the length of the time step set as a parameter.

One of the standard methodologies to perform these inversions is based on solving a system of equations given by the observed profile and the combination of Equations (1) and (3) (Vasseur et al., 1983; Beltrami et al., 1992; Shen et al., 1992; Hartmann and Rath, 2005). This approach uses a Singular Value Decomposition (SVD) algorithm to solve this system, and it is known as





SVD inversion method (Figure 1). The system considered in this inversion method can be expressed as

$$\mathbf{T}_{obs} = \mathbf{M}\mathbf{T}_{model}, \tag{4}$$

where $\mathbf{T}_{obs}$ is the anomaly temperature profile, $\mathbf{T}_{model}$ is a vector containing the step change model of the surface temperature to be determined, and $\mathbf{M}$ is the matrix containing the coefficients given by

$$M_{i,j} = \mathrm{erfc}\left(\frac{z_i}{2\sqrt{\kappa t_j}}\right) - \mathrm{erfc}\left(\frac{z_i}{2\sqrt{\kappa t_{j-1}}}\right). \tag{5}$$

This system of equations, nevertheless, is overdetermined. This means that there are more equations in the system than parameters to be determined, thus the solution is non-unique. A Singular Value Decomposition (SVD) algorithm (Lanczos, 1961) has been extensively used to solve these overdetermined systems (e.g., Mareschal and Beltrami, 1992; Clauser and Mareschal, 1995; Jaume-Santero et al., 2016). The SVD algorithm is based on decomposing the matrix of coefficients $\mathbf{M}$ into two orthogonal matrices ($\mathbf{U}$ and $\mathbf{V}$) and a rectangular matrix ($\mathbf{S}$) containing the eigenvalues in the diagonal:

$$\mathbf{M} = \mathbf{U}\mathbf{S}\mathbf{V}^T. \tag{6}$$

Thereby, the solution of the system can be found as

$$\mathbf{T}_{model} = \mathbf{V}\mathbf{S}^{-1}\mathbf{U}^T\mathbf{T}_{obs}. \tag{7}$$

Finally, small eigenvalues are removed from $\mathbf{S}^{-1}$ for stabilizing the solution, but at the cost of losing temporal resolution in the model. The final number of eigenvalues used to retrieve the solution is an important parameter, retaining typically two or three eigenvalues. More details about SVD inversions can be found in the literature (Mareschal and Beltrami, 1992; Clauser and Mareschal, 1995; Beltrami, 2001; González-Rouco et al., 2009; Cuesta-Valero et al., 2021c).

The SVD inversion (Figure 2) is applied to the anomaly profiles, that is, the resulting perturbation profiles after removing the quasi-equilibrium profile from the measured log (Figures 1b, c). The quasi-equilibrium profile, in other words the term $T_0 + \Gamma_0 \cdot z$ in Equation (1), is estimated using a linear regression analysis of the deepest $100$ m of the corresponding profile, which is the part of the measured profile not affected by recent changes in surface conditions. The estimated quasi-equilibrium profile is then subtracted from each STP. The error in the estimates of the long-term surface temperature ($T_0$) and the geothermal gradient ($\Gamma_0$) has been typically included in the analysis by inverting three anomaly profiles, the anomaly profile resulting of subtracting the best estimate of $T_0$ and $\Gamma_0$ from the measured profile (black dots in Figure 1b), and two extremal profiles resulting of subtracting the quasi-equilibrium profiles determined by the errors in $T_0$ and $\Gamma_0$ (red and blue dots in Figure 1b). These extremal profiles are derived by subtracting and adding the corresponding two sigma values to the best estimate of $T_0$ and $\Gamma_0$. The inversions of the extremal anomaly profiles constitute the upper and lower boundaries of the uncertainty range (Beltrami et al., 2015a, b) in the retrieved ground surface temperature history of the corresponding log (red and blue lines in Figure 1a).

Ground heat flux histories for each STP are estimated from ground surface temperature histories (see Section 3.6). Concretely, a flux history is estimated from each temperature history retrieved from each STP, that is, from the best estimate and





the inversion of the two extremal anomaly profiles. Flux histories can also be retrieved from flux profiles, which can be esti-
mated from the measured STPs by applying the Fourier equation (Beltrami, 2001; Cuesta-Valero et al., 2021c). Inversions of
flux profiles are, nevertheless, noisier than inversions of temperature profiles, thus we decided to derive heat flux histories from
temperature histories in this analysis.

Please note that the method described in this section is applied to individual STPs, see Section 3.7 for a description of the
approach used to obtain results from a set of profiles.

## 3.4 Perturbed Parameter Inversions

Inverting the extremal anomaly profiles obtained from the errors in estimating $T_0$ and $\Gamma_0$ allows to account for the uncertainty
in the determination of the quasi-equilibrium temperature profile in the SVD approach described above. Nevertheless, the SVD
approach is not able to include the uncertainty due to the unknown thermal properties at each site. Hence, a more comprehensive
way to estimate uncertainties in STP inversions was developed in Cuesta-Valero et al. (2021c). This approach, called Perturbed
Parameter Inversion (PPI), consists in generating a large ensemble of SVD inversions from each subsurface temperature profile
by varying the values of the inversion parameters: time step of the surface signal, thermal diffusivity and conductivity, and the
number of eigenvalues retained in Equation (7) (Figure 2). This process is repeated for the three anomaly profiles retrieved to
characterize the uncertainty in the determination of the quasi-equilibrium temperature profile (see SVD technique above). We
perturb only the parameters related to thermal properties and the quasi-equilibrium temperature profile in this study, as these are
the most important sources of uncertainty (Cuesta-Valero et al., 2021c). That is, we generate an ensemble of inversions using
the three anomaly profiles obtained from the best estimate and two sigma values of the intercept ($T_0$) and slope ($\Gamma_0$) determined
from the linear regression of the deepest part of the profile (see above for the SVD case), and considering a different thermal
diffusivity for each inversion ($N_{Diffusivity}$ in Figure 2). Thereby, the ensemble contains $3 \times N_{Diffusivity}$ elements. The 2.5th,
50th, and 97.5th percentiles of all retrieved temperature histories are then estimated in order to provide with a best estimate
(the median) and a 95% confidence interval for the ground surface temperature history of each profile.

The PPI method described in Cuesta-Valero et al. (2021c) removed highly unlikely individual ground surface temperature
histories from the final ensemble. However, we consider here all possible histories in order to explore the full range of variability
in the inversions, including unlikely cases. For the same reason, forward models of individual histories are not compared with
the anomaly profiles, and thus each history is weighted equal in the estimated percentiles, in contrast to the approach in Cuesta-
Valero et al. (2021c). Thereby, the full effect of unknown thermal properties in the estimated uncertainty can be assessed by
comparing PPI solutions with SVD inversions, since the only difference between these two techniques is the use of a range of
diffusivities in the PPI method.

Ground heat flux histories for each STP are estimated from ground surface temperature histories (see Section 3.6) similarly
to the SVD case, but with some differences. The PPI method retrives an ensemble of temperature histories for each STP. We
estimate another ensemble containing flux histories using each of the temperature histories in the PPI ensemble, obtaining the
2.5th, 50th, and 97.5th percentiles that constitute the best estimate and uncertainty range for the ground heat flux history of the
corresponding profile.





As in the case of the description of the SVD technique, note that this method is applied to individual STPs, see Section 3.7 for a description of the approach used to obtain results from a set of profiles.

## 3.5   Bootstrap Sampling Strategy

As explained in the Introduction, the SVD and PPI techniques do not provide with a comprehensive statistical framework to estimate the uncertainty resulting from aggregating inversions from several STPs. In order to overcome this problem, we have developed a new method to retrieve ground surface temperature histories combining an SVD algorithm and a bootstrapping sampling strategy that provides with a meaningful statistical method to estimate uncertainty ranges for aggregations of any number of individual profiles (Efron, 1987; DiCiccio and Efron, 1996; Davison and Hinkley, 1997).

Bootstrap Inversions (BTIs) are based on generating two ensembles constituted by inversions performed by the SVD algorithm, named Sampling and Bootstrapping ensembles (S and B ensembles in Figure 2). The Sampling ensemble consists in SVD inversions of an individual log or a population of logs with parameters randomly selected from a range of possibilities. The BTI method considers the uncertainty arising from determining the quasi-equilibrium profile (i.e., the errors in $T_0$ and $\Gamma_0$), and the unknown thermal diffusivity and conductivity (Table 1). All possible values for each parameter are equally probable, except for the quasi-equilibrium temperature profile. In this case, the long-term surface temperature and geothermal gradient are given by a Gaussian distribution with mean and standard deviation corresponding to the best estimate and error for $T_0$ and $\Gamma_0$, which are retrieved form the linear regression analysis of the deepest part of each profile (see above). Therefore, the Sampling ensemble includes an inversion from each individual log with random parameters when considering several profiles, and just an inversion with random parameters when considering one profile, thus the size of the Sampling ensemble corresponds to the number of profiles used.

A number of different Sampling ensembles are created, with the mean inversion of each Sampling ensemble constituting an element of the Bootstrapping ensemble (Figure 2). Here we consider 1,000 different Sampling ensembles, thus the Bootstrapping ensemble includes 1,000 averaged inversions. Larger sizes for the Bootstrapping ensemble can be considered, but the results are approximately the same (see analysis below). The median of the Bootstrapping ensemble constitutes the best estimate for the mean ground surface temperature history, with the 95% confidence interval given by the 2.5th and 97.5th percentiles of the Bootstrapping ensemble (Efron, 1987; DiCiccio and Efron, 1996; Davison and Hinkley, 1997).

As for the SVD and PPI cases, ground heat flux histories are derived from ground surface temperature histories for each STP. Each temperature history considered in the Sampling ensemble is used to estimate a flux history, which is then averaged to obtain a global mean ground heat flux history. The process is repeated 1,000 times as in the case of temperature histories to generate a Bootstrapping ensemble containing mean flux histories, from which the 2.5th, 50th, and 97.5th percentiles are computed.

## 3.6   Ground Heat Flux Histories

Ground heat flux histories are estimated using the method developed in Wang and Bras (1999) using a half-order derivative approach to estimate ground heat flux from soil temperatures in a homogeneous medium. Since SVD inversions also assume





a homogeneous subsurface, this method can be applied to ground surface temperature histories to obtain ground heat flux histories (Beltrami et al., 2002; Cuesta-Valero et al., 2021c, among others). Given a temperature history with equidistant time steps ($t_N$), the corresponding heat flux history is estimated as

$$G(t_N) = \frac{2\lambda}{\sqrt{\pi\kappa\Delta t}} \sum_{i=1}^{N-1} \left[ (T_{i+1} - T_i)\left(\sqrt{N-i} - \sqrt{N-i+1}\right) \right], \tag{8}$$

285  where, $G$ is ground heat flux, $\kappa$ is thermal diffusivity, $\lambda$ is thermal conductivity, $\Delta T$ is the size of the time step, and $T_i$ is the value of the temperature history at time step $i$. This equation is applied to ground surface temperature histories retrieved from SVD, PPI, and BTI techniques to estimate ground heat flux histories, with a range of plausible values for thermal conductivity and thermal diffusivity in the BTI case (Table 1).

### 3.7 Global Averages

290  Global estimates of ground surface temperature from individual Xibalbá STPs are retrieved using the SVD, PPI, and BTI techniques. Global estimates from individual SVD inversions are obtained by averaging temperature and flux histories from each individual log, while the global uncertainty corresponds to the mean of the inversions of the extremal anomaly profiles. That is, the two extremal anomaly profiles derived by subtracting and adding the two sigma values of $T_0$ and $\Gamma_0$ to the best estimate of these parameters are inverted for each profile, constituting the upper and lower limits of the uncertainty range for

295  the ground surface temperature history retrieved using the best estimate of $T_0$ and $\Gamma_0$ (i.e., the red and blue lines in Figure 1). The mean of all the upper limits (red line) and the mean of all the lower limits (blue line) estimated from the 1,079 Xibalbá profiles constitute the uncertainty range for the global average. A similar approach is applied in the case of individual PPI solutions, but averaging the 2.5th, 50th, and 97.5th percentiles of each STP in the Xibalbá dataset. That is, the mean of all 50th percentiles is considered to be the global averaged history, and the interval given by the mean of all 2.5th and all

300  97.5th percentiles is considered to be the corresponding uncertainty range. This approach has been used in previous studies to derive global estimates of temperature and heat flux changes in the land surface from SVD and PPI inversions (Beltrami et al., 2015a, b; Cuesta-Valero et al., 2021c), although this is not a meaningful statistical method. As previously discussed, the BTI framework can yield results from any number of logs by modifying the size of the Sampling ensemble. Therefore, global averages of temperature histories are estimated by including an inversion from each of the 1,079 Xibalbá STPs in the Sampling

305  ensemble. The mean of this 1,079 inversions constitutes one member of the Bootstrapping ensemble, which therefore consists in 1,000 global averages each one estimated from 1,079 different inversions, one from each profile in the dataset. The final global results are obtained by computing the 2.5th, 50th, and 97.5th percentiles of the Bootstrapping ensemble.

The same approaches are applied to ground heat flux histories from Xibalbá profiles to derive global averaged heat flux histories.





## 3.8 Inversion Parameters

We apply the SVD, PPI, and BTI techniques described above to an artificial subsurface anomaly profile generated from the PAGES2k-Land temperatures (Figure 1a) and to the 1,079 world-wide subsurface temperature profiles included in the Xibalbá database. We consider a range of possible values for each relevant parameter (see Table 1) in order to determine the uncertainty rising from poorly known quantities in the inversions.

The continental subsurface is considered as homogeneous for all inversions performed here, thus we assume no variation of the diffusivity or conductivity with depth. For SVD inversions, we use a constant thermal diffusivity of $1 \times 10^{-6}$ $\mathrm{m^2\,s}$, which is a typical value for these type of inversions (e.g., Hartmann and Rath, 2005; Jaume-Santero et al., 2016; Pickler et al., 2016, 2018). Thermal conductivity is also considered as constant through the subsurface, with a value of $3\,\mathrm{W\,m^{-1}\,K^{-1}}$ that also matches the conductivity used in other works based on SVD inversions (Cermak and Rybach, 1982; Beltrami, 2001; Beltrami et al., 2002; Beltrami, 2002; Cuesta-Valero et al., 2021c). For the PPI and BTI approaches, we consider a range of thermal diffusivities from $0.5 \times 10^{-6}$ $\mathrm{m^2\,s}$ to $1.5 \times 10^{-6}$ $\mathrm{m^2\,s}$. The BTI technique also takes into account a range of thermal conductivities from $2.5\,\mathrm{W\,m^{-1}\,K^{-1}}$ to $3.5\,\mathrm{W\,m^{-1}\,K^{-1}}$ to estimate ground heat fluxes from temperature histories (Equation 8). Nevertheless, the PPI approach considers a constant thermal conductivity of $3\,\mathrm{W\,m^{-1}\,K^{-1}}$ for estimating flux histories in order to simplify the comparison with results from SVD inversions. The considered ranges for thermal diffusivity and thermal conductivity contain all plausible values addressed by the literature (Shen et al., 1995; Harris and Chapman, 2005; Hartmann and Rath, 2005; Woodbury and Ferguson, 2006; Chouinard et al., 2007; Hopcroft et al., 2007; Huang et al., 2008; Hopcroft et al., 2009; Davis et al., 2010; Rath et al., 2012; Demezhko and Gornostaeva, 2015b; Burton-Johnson et al., 2020).

All SVD, PPI and BTI inversions use the same model for the boundary condition, i.e., the temporal signal at the surface, consisting in a series of step changes all with the same temporal length. We perform inversions with time steps of 10, 30, and 50 years to test the effect of this parameter on the results produced by the three techniques. We also obtain SVD, PPI, and BTI inversions retaining the two and three largest eigenvalues in the decomposition matrix (Equation 7) in order to test the dependency of the retrieved histories on this parameter.

## 4 Results

### 4.1 Methodological Evaluation

An artificial profile generated from a known surface temperature allows to evaluate the ability of the SVD, PPI, and BTI techniques described above to retrieve the original surface signal under perfect conditions (Figure 1). Ground surface temperature histories obtained by applying the SVD and PPI methodologies recover the main features of the PAGES2k-Land temperatures used as surface boundary condition to generate the artificial profile (purple and red lines in Figure 3). Inversions using two and three eigenvalues present similar temperature histories, as well as similar uncertainty ranges, except at the end of the period when inversions performed with 10-year step changes in the surface signal with two eigenvalues yield larger uncertainties than inversions with three eigenvalues. Bootstrap inversions show similar temperature histories than those from SVD and PPI





inversions, with uncertainty ranges slightly larger than those from SVD inversions and slightly lower than those from PPI inversions (Figure 3). This is an expected result, as the BTI approach considers a range of possible values for thermal diffusivity, while SVD inversions consider only one value for diffusivity (Table 1). Also, PPI uncertainty ranges are expected to be larger

than BTI confidence intervals because the PPI approach considers the two sigma values in long-term surface temperature and geothermal gradient for determining the two extremal anomaly profiles (red and blue dots in Figure 1b). That is, the extremal anomalies in the PPI method are always going to represent a highly improbable case of the Gaussian distribution of possible values for long-term surface temperature and geothermal gradient, while the BTI approach samples randomly the possible anomalies given by a Gaussian distribution around the regression line obtained to estimate the quasi-equilibrium profile, thus

the 2.5th and 97.5th percentiles for the BTI inversions are always contained in the inversions of the extremal anomalies in the PPI case.

Small differences in ground surface temperature histories are present in 30-year (Figure 3c) and 50-year (Figures 3e) time step inversions retaining the two largest eigenvalues in the solution (Equation 7). More importantly, the averaged ground surface temperature histories from inversions with time steps of 50-years and three eigenvalues show negative temperature changes

around 1925 CE in SVD, PPI, and BTI cases, while the original surface temperature is increasing during the same period (Figure 3f). This suggests that the third eigenvalue may be excessively small, leading to an unrealistic level of variability in the solutions of Equation (7).

Results from the new bootstrapping technique generally recover the original surface temperature used to derive the artificial profile (Figure 3). The ground surface temperature histories retrieved by the BTI method, nevertheless, present a new parameter

affecting the behaviour of the retrieved histories, the size of the Bootstrapping ensemble (see Figure 2, Table 1). The bootstrap approach is based on randomly sampling with repetition a number of different populations from the original, finite data (Efron, 1987; DiCiccio and Efron, 1996; Davison and Hinkley, 1997). An estimate of the probability density function for the desired statistic is then retrieved from these different populations, but the number of populations (i.e., the size of the Bootstrapping ensemble) is going to determine the final results. In the case of the retrieved temperature history from the artificial profile,

we used a Bootstrapping ensemble with 1,000 members, and an increase in the number of populations considered does not change the results (Figures 4a). The retrieved uncertainty range is also sensible to the size of the Bootstrapping ensemble, but considering more than 1,000 populations does not change much the estimated uncertainty (Figure 4c).

### 4.2 Inverting the Xibalbá Database

Inversions of the artificial profile have shown that the SVD, PPI, and BTI methods are able to yield very similar averaged

ground surface temperature histories for one log, in agreement with the prescribed surface temperatures. However, this is a theoretical case, and real case applications would involve more complex situations, including a higher number of profiles with more noisy temperature records. This section describes the results obtained from the application of these three techniques to the subsurface temperature profiles from the Xibalbá dataset, including the comparison with global land temperatures estimated from PAGES2k multi-proxy reconstructions and previous estimates from Cuesta-Valero et al. (2021c). Global ground heat flux

histories are also retrieved and compared with estimates from Cuesta-Valero et al. (2021c).





Global averaged ground surface temperature histories from the Xibalbá dataset retrieved from the SVD, PPI, and BTI techniques are very similar for inversions considering time steps of 10, 30, and 50 years for the surface history, and for inversions retaining the largest two and three eigenvalues (Figure 5). The most important difference between inversion methods consists in the estimated uncertainty ranges, with PPI results yielding the broadest interval and BTI inversions the narrowest interval.

In fact, the uncertainty range of the bootstrap approach is much smaller than that from SVD and PPI inversions for the global case, in contrast with the similar values achieved by the three techniques for the artificial profile (Figure 3). This is caused by the differences in the methods used to estimate global uncertainties in the three approaches (see Section 3.7). The SVD and PPI methods aggregate the uncertainty of individual profiles to derive global uncertainty estimates. Concretely, these approaches average the upper and lower bounds of the individual uncertainty ranges, which provides with a conservative global uncertainty

range. That is, it is highly probable that the global averaged ground surface temperature history is contained in this interval because it is markedly broad by construction. Nevertheless, this uncertainty range is not interpretable statistically, thus hindering its assessment against other past temperature estimates from STP inversions using Bayes methods, climate simulations, or proxy-based reconstructions. In contrast, the bootstrapping approach provides with a meaningful statistical framework to derive global uncertainties. That is, the global uncertainty range retrieved by the BTI technique can be interpreted as the 95%

confidence interval of the global averaged ground surface temperature history, as it considers different possible values of the global average estimated using different, but possible, values for the unknown parameters in the inversion. Therefore, results from the SVD and PPI techniques are overestimating the global uncertainty due to an inadequate aggregation of individual uncertainties.

As in the case of the artificial profile, BTI results are sensible to the number of resamplings considered, i.e., the size of

the Bootstrapping ensemble (Figure 2). The dependency on the size of the Bootstrapping ensemble of the global averaged temperature history and the global uncertainty range from Xibalbá profiles converges to a stable value when the number of samplings increases, with small changes after considering a Bootstrapping ensemble of 1,000 members (Figure 4b and c). Considering more than 1,000 ensemble members, therefore, does not change much the results, thus 1,000 is the recommended size for the Bootstrapping ensemble when applying the BTI technique to individual profiles and large sets of profiles, given the

results of Figure 4.

Global averaged ground heat flux histories estimated from Xibalbá profiles show similar results using the SVD, PPI, and BTI methods (Figure 6). There is an unexpected decrease in the heat flux at the end of the 20[th] century for solutions considering 10-year step changes at the surface (Figure 6a, b), which is not present in solutions with 30-year and 50-year time steps, nor in previous estimates of global heat flux histories (Beltrami et al., 2002; Beltrami, 2002; Cuesta-Valero et al., 2021c).

Furthermore, this flux decrease is present in all three methods retaining the two and three largest eigenvalues in the inversions, indicating that a 10-year step change in the model for surface boundary conditions may be excursively small, at least to retrieve past histories of ground heat flux. There are also differences in the retrieved uncertainty range for the global mean from each technique, with BTI results displaying a much smaller uncertainty than SVD and PPI results. This is an expected result, since the ground heat flux histories from the three inversion methods are estimated from ground surface temperature histories, which

also yielded these differences in the retrieved uncertainty range (Figure 5).





The assessment of ground surface temperature histories retrieved from Xibalbá STPs using the SVD, PPI, and BTI methods shows agreement with previous estimates from the same logs retrieved with a slightly different PPI technique (Figure 7a and Table 2). The main methodological difference between the PPI approach used here and the approach used in Cuesta-Valero et al. (2021c) consists in the filtering of individual inversions to remove unrealistic histories in comparison with long-term changes in meteorological data. This additional screening removed the histories showing the largest variability with time, reducing the uncertainty range in the estimates of ground heat flux histories and ground surface temperature histories. Therefore, the lack of filtering in the PPI approach applied here explains the smaller uncertainty range in Cuesta-Valero et al. (2021c). Another remarkable result is the agreement between the PAGES2k land temperatures and the ground surface temperature histories for most of the period. Both PAGES2k-Land temperatures and surface temperature histories from Xibalbá logs display stable conditions for the period 1600-1800 CE, with temperature histories warming rapidly during 1800-2000 CE and PAGES2k temperatures slightly deceasing between 1800-1900 CE and increasing from 1900 to 2000 CE. As result of this different evolution in the period 1800-1900 CE, PAGES2k data present lower temperature changes in comparison with preindustrial conditions during the 20th century than ground surface temperature histories, but similar temperature trends (Table 2). This difference in warming from multi-proxy data and subsurface temperature profiles is a known problem that needs further investigation in order to reconcile the estimated temperature evolution during the last millennium from both datasets. Nevertheless, results in Figure 7a show smaller differences than expected in comparison with previous analyses (Huang et al., 2000; Harris and Chapman, 2005; Jaume-Santero et al., 2016; Beltrami et al., 2017). PAGES2k-Land temperatures have been horizontally displaced to have the same period of reference as the STP inversions, that is, 1300-1700 CE. Therefore, the small offset around 1600 CE is just caused by the natural variability of the PAGES2k series, which includes higher-frequency signals than STP results.

The averaged ground heat flux histories derived here are also in agreement with those from Cuesta-Valero et al. (2021c) despite several methodological differences (Figure 7b). As indicated above, the PPI method in Cuesta-Valero et al. (2021c) removed individual inversions with high variability from the PPI ensemble, sampled a range of possible conductivities, and weighted each inversion in comparison with the measured profile. This three factors, particularly the sampling of a range of conductivities, lead to higher uncertainty in Cuesta-Valero et al. (2021c) than in the PPI ground heat flux histories retrieved here.

## 5  Discussion

The new BTI technique, based on combining a broadly used SVD algorithm with a bootstrap sampling approach, presents robust estimates of ground temperature histories and ground heat flux histories in comparison with results from the SVD and PPI methods, both for an artificial case and for real STPs. A number of possible values for several inversion parameters is considered, including different lengths of the time steps for retrieving the surface histories, and the number of eigenvalues retained in the inversion, with all cases showing BTI results in agreement with the SVD and PPI techniques. The effect of considering different lengths for the time steps to retrieve surface histories is small, consisting in lower temperature changes with increasing time steps, since solutions retrieve the averaged surface signal for a longer period of time. Nevertheless,





there are some unrealistic results when considering 10-year time steps and when retaining the three largest eigenvalues in the
solution. Therefore, inversions performed with around 30-year and 50-year time steps retaining the two largest eigenvalues
in the solutions are more adequate for inverting STPs from the Xibalbá global dataset, but the time step and the number of
retained eigenvalues should be selected on a case-by-case basis, depending on the application of interest.

The most important difference of the new bootstrapping technique with previous methods arises when aggregating the re-
trieved inversions to estimate the global mean temperature history and the global mean heat flux history, as the uncertainty
range derived in this case is markedly smaller for BTI results than for SVD and PPI results. The SVD and PPI techniques
average the upper and lower limits of the uncertainty range of all profiles (red and blue lines in Figure 1a) to provide with
an estimate of the global uncertainty, while the bootstrap technique derives a high number (1,000 in this study) of different
global means, from which the 2.5th and 97.5th percentiles are estimated. In other words, the global uncertainties from the
SVD and PPI techniques are conceptually different from the uncertainty estimated from the BTI technique. In fact, the SVD
and PPI uncertainties are difficult to interpreted using standard statistical paradigms, while the uncertainty estimates of the
BTI approach follow the general principles of bootstrapping. However, the global uncertainty from the SVD and PPI methods
converges to that of the bootstrap approach if considering the uncertainty from the inversion of individual logs as Gaussian
errors and applying standard error propagation (Figure 8). That is, if uncertainty in SVD and PPI inversions from individual
profiles is considered to be Gaussian, the standard error of each individual log can be used to derive a standard error for the
global mean using typical error propagation methods, instead of averaging the limits of the uncertainty range. Figure 8 displays
the uncertainty ranges for the global averaged temperature and heat flux histories estimated by standard error propagation of
individual SVD and PPI results, as well as the confidence interval from the bootstrapping approach. The uncertainty ranges
from the three techniques present very similar values for the entire period of interest when considering a common value of
thermal diffusivity. Small differences in the retrieved uncertainty ranges arise when considering different possible values for
the unknown thermal properties at each location, that is, when the PPI and BTI techniques include a range of possible values
for thermal properties. However, the retrieved errors in individual SVD and PPI inversions do not follow, in general, a Gaus-
sian distribution (see uncertainty ranges in Figure 3), thus the use of error propagation with the SVD and PPI solutions is not
advised. The BTI method, nevertheless, do not make any assumption regarding the distribution of random errors, which is an
advantage for aggregating individual inversions. Another factor leading to the small confidence intervals for the global mean
surface temperature and heat flux histories achieved by the BTI technique is the lack of high-frequency signals in inversions
of subsurface temperature profiles. As explained in the Methods section, the ground acts as a filter removing the short-period
alterations in the surface energy balance, thus measured STPs only retain long-term changes in surface conditions (e.g., Smer-
don and Stieglitz, 2006). Thereby, STP inversions contain only long-term variability, reducing the uncertainty in the retrieved
surface histories. Overall, we conclude that the bootstrap technique is a more adequate method to aggregate results from indi-
vidual subsurface temperature profiles from the point of view of Statistics, and it provides more interpretable and meaningful
uncertainty estimates for global averages than the typical SVD and PPI approaches.

The similar global histories of ground surface temperature and ground heat flux for the three methods assessed here indicate
that the ground component of the Earth heat inventory and the estimated temperature change since preindustrial times from STP





inversions using the bootstrap method are also similar to previous results. Ground heat flux from the bootstrapping technique

yields $67.2 \, \mathrm{W \, m^{-2}}$ for the period 1950-2000 CE (Table 2), which correspond to a global ground heat storage of $14 \times 10^{21}$ J, in agreement with the results of von Schuckmann et al. (2020). The global temperature change since preindustrial times, around 1300-1700 CE in this case (Cuesta-Valero et al., 2019), can be estimated by assuming that land surface temperature changes are around 2.36 times larger than sea surface temperature changes (Harrison et al., 2015), thus obtaining an increase of $0.7 \, ^{\circ}\mathrm{C}$ in the global temperature. That is the same estimate as in Cuesta-Valero et al. (2021c) using Xibalbá profiles. However, the

uncertainty in estimates performed using the BTI technique is smaller than in previous results by an order of magnitude (Table 2). This is relevant to assess the total uncertainty in the Earth heat inventory and to compare with other components of the continental heat storage, such as inland water bodies (Vanderkelen et al., 2020). Furthermore, the uncertainty range of the temperature changes since preindustrial times from subsurface temperature profiles is now smaller than the uncertainty in estimates from other sources ($0.6 - 0.8 \, ^{\circ}\mathrm{C}$, Hawkins et al., 2017).

**6   Conclusions**

A new bootstrap technique to quantify uncertainties in SVD inversions of subsurface temperature profiles has been described and tested, obtaining robust results in comparison with other methods. This new technique reaches similar values of ground heat storage and temperature change since preindustrial times to previous studies, although providing with more meaningful uncertainty ranges. The bootstrap method is able to incorporate several important sources of uncertainty affecting the inversion

of subsurface temperature profiles in a flexible framework that allows the analysis of individual profiles as well as the analysis of datasets with thousands of profiles, and it is based on robust statistical paradigms.

The new bootstrap approach estimates lower uncertainties for global histories of surface temperature and heat flux than other inversion methods. This result reinforces the role of inversions of subsurface temperature profiles as indicators of long-term changes in surface conditions, thus the importance of expanding the global network of profiles to include measurements in the

southern hemisphere, the Middle East and southeastern Asia, as well as to include more recent measurements at previously sampled sites in order to include the recent land warming in global estimates from inversions of subsurface temperature profiles.

*Code availability.* The CIBOR 1.0 collection of scripts will be available at Zenodo after the manuscript is accepted.

*Data availability.* Xibalbá temperature profiles are available from Cuesta-Valero et al. (2021a). This dataset will be updated to include individual and global inversions performed with the SVD, PPI, and BTI techniques after the manuscript is accepted. Multi-proxy temperatures

were retrieved from the data associated to the Summary for Policymakers of the sixth Assessment Report of the IPCC (Gillett et al., 2021; IPCC, 2021).





*Author contributions.* F.J.C.V. wrote the manuscript with continuous feedback from all authors. All authors contributed to the design of this study.

*Competing interests.* The authors declare no competing interest.

*Acknowledgements.* F.J.C.V is an Alexander von Humboldt Research Fellow at the Helmholtz Centre for Environmental Research (UFZ). H.B. holds the Canada Research Chair in Climate Dynamics, and S.G. holds the Canada Research Chair in Climate Change Impacts/Adaptation in Northern Canada. This work was partially supported by grants from the Natural Sciences and Engineering Research Council of Canada Discovery Grants (NSERC DG 140576948 and 2020-04783). Part of the presented analysis was performed in the computational facilities provided by the Atlantic Computational Excellence Network (ACENET-Compute Canada). This analysis contributes to the PALEOLINK project
(https://www.pastglobalchanges.org/science/wg/2k-network/projects/paleolink/intro, last access: February 9th, 2022), part of the PAGES 2k Network.





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





**Table 1.** Summary of the values for inversion parameters used to invert the artificial profile and logs from Xibalbá dataset. Columns contain the technique employed, the length of the time step used to retrieve the surface histories, the number of eigenvalues retrieved in the inversions, the values of thermal diffusivity considered, the values of thermal conductivity used to derive ground heat flux histories, the size of the Sampling ensemble of the BTI approach, and the size of the Bootstrapping ensemble of the BTI approach. All combination of parameters are explored in each case.

| Technique | Step Change | Eigenvalues | Diffusivity | Conductivity | $N_S$ | $N_B$ |
|---|---|---|---|---|---|---|
| Artificial Profile | | | | | | |
| SVD | 10, 30, 50 years | 2, 3 | $1 \times 10^{-6}$ m$^2$ s | - | - | - |
| PPI | 10, 30, 50 years | 2, 3 | From $0.5 \times 10^{-6}$ m$^2$ s to $1.5 \times 10^{-6}$ m$^2$ s (N = 100) | - | - | - |
| BTI | 10, 30, 50 years | 2, 3 | From $0.5 \times 10^{-6}$ m$^2$ s to $1.5 \times 10^{-6}$ m$^2$ s (N = 1,000) | - | 1 | 10, 100, 1,000, 10,000, 100,000, 1,000,000 |
| Xibalbá Dataset | | | | | | |
| SVD | 10, 30, 50 years | 2, 3 | $1 \times 10^{-6}$ m$^2$ s | 3 W m$^{-1}$ K$^{-1}$ | - | - |
| PPI | 10, 30, 50 years | 2, 3 | From $0.5 \times 10^{-6}$ m$^2$ s to $1.5 \times 10^{-6}$ m$^2$ s (N = 100) | 3 W m$^{-1}$ K$^{-1}$ | - | - |
| BTI | 10, 30, 50 years | 2, 3 | From $0.5 \times 10^{-6}$ m$^2$ s to $1.5 \times 10^{-6}$ m$^2$ s (N = 1,000) | From 2.5 W m$^{-1}$ K$^{-1}$ to 3.3 W m$^{-1}$ K$^{-1}$ (N = 1,000) | 1,079 | 10, 100, 1,000, 10,000 |

**Table 2.** Averaged ground surface temperature histories and averaged ground heat flux histories retrieved using the SVD, PPI and BTI techniques described in the Methods section. Temperature changes from the PAGES2k-Land temperatures and temperature and flux histories from Cuesta-Valero et al. (2021c) (CV21) are also shown for comparison purposes. Results from the BTI technique consider $N_S = 1,079$ and $N_B = 1,000$. Temperatures in °C, fluxes in W m$^{-2}$.

| | Temperature | | | | | Heat Flux | | | |
|---|---|---|---|---|---|---|---|---|---|
| Period CE | PAGES2k | CV21 | SVD | PPI | BTI | CV21 | SVD | PPI | BTI |
| 1950-2000 | $0.57 \pm 0.18$ | $1.00 \pm 0.25$ | $1.01 \pm 0.30$ | $1.0 \pm 0.5$ | $1.008 \pm 0.025$ | $0.065 \pm 0.033$ | $0.068 \pm 0.005$ | $0.067 \pm 0.022$ | $0.0672 \pm 0.0028$ |
| 1900-1950 | $0.22 \pm 0.19$ | $0.5 \pm 0.5$ | $0.5 \pm 0.5$ | $0.5 \pm 0.8$ | $0.50 \pm 0.04$ | $0.03 \pm 0.04$ | $0.030 \pm 0.013$ | $0.029 \pm 0.023$ | $0.0318 \pm 0.0018$ |
| 1850-1900 | $-0.00 \pm 0.20$ | $0.2 \pm 0.4$ | $0.2 \pm 0.5$ | $0.2 \pm 0.9$ | $0.22 \pm 0.04$ | $0.012 \pm 0.030$ | $0.012 \pm 0.015$ | $0.011 \pm 0.022$ | $0.0143 \pm 0.0014$ |
| 1800-1850 | $-0.07 \pm 0.21$ | $0.09 \pm 0.29$ | $0.1 \pm 0.4$ | $0.1 \pm 0.7$ | $0.110 \pm 0.032$ | $0.008 \pm 0.020$ | $0.005 \pm 0.011$ | $0.005 \pm 0.018$ | $0.0071 \pm 0.0011$ |
| 1750-1800 | $0.03 \pm 0.24$ | $0.03 \pm 0.26$ | $0.03 \pm 0.33$ | $0.0 \pm 0.6$ | $0.057 \pm 0.026$ | $0.005 \pm 0.022$ | $0.003 \pm 0.008$ | $0.002 \pm 0.014$ | $0.0039 \pm 0.0008$ |
| 1700-1750 | $-0.06 \pm 0.27$ | $-0.01 \pm 0.25$ | $0.01 \pm 0.28$ | $0.0 \pm 0.5$ | $0.030 \pm 0.022$ | $0.004 \pm 0.024$ | $0.001 \pm 0.006$ | $0.001 \pm 0.011$ | $0.0023 \pm 0.0006$ |
| 1650-1700 | $-0.06 \pm 0.34$ | $-0.03 \pm 0.25$ | $-0.00 \pm 0.23$ | $-0.0 \pm 0.5$ | $0.014 \pm 0.019$ | $0.003 \pm 0.025$ | $0.001 \pm 0.004$ | $0.001 \pm 0.008$ | $0.0013 \pm 0.0004$ |
| 1600-1650 | $-0.1 \pm 0.4$ | $-0.04 \pm 0.24$ | $-0.01 \pm 0.20$ | $-0.0 \pm 0.4$ | $0.005 \pm 0.017$ | $0.003 \pm 0.025$ | $0.0004 \pm 0.0031$ | $0.000 \pm 0.006$ | $0.00078 \pm 0.00031$ |

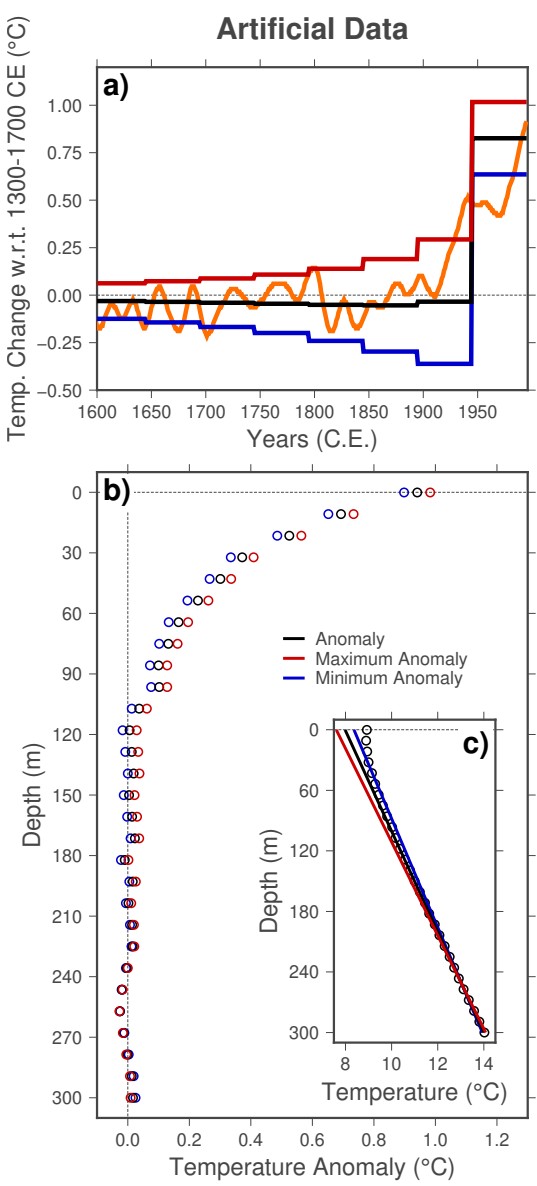

**Figure 1.** (a) Surface signal (orange line), and corresponding anomaly (b) and original (c) profiles for the artificial case. The surface signal corresponds to the PAGES2k-Land temperatures described in the Data section. The subsurface temperature profiles result from the propagation of the surface signal through the ground using a purely conductive forward model (see Methods section), considering a quasi-equilibrium profile given by a long-term surface temperature of $T_0 = 8\,°C$ and a geothermal gradient of $\Gamma = 0.02\,°C\,m^{-1}$. Gaussian noise with $\mu = 0.0\,°C$ and $\sigma = 0.02\,°C$ is added to simulate measurement error. The extremal profiles (red and blue data) are estimated from a linear regression analysis of the deeper 100m of the profile. The intercept of the extremal profiles in panel (c) has been modified for clarity. Temperature reconstructions from SVD inversions of each anomaly profile in panel (b) are shown in panel (a).



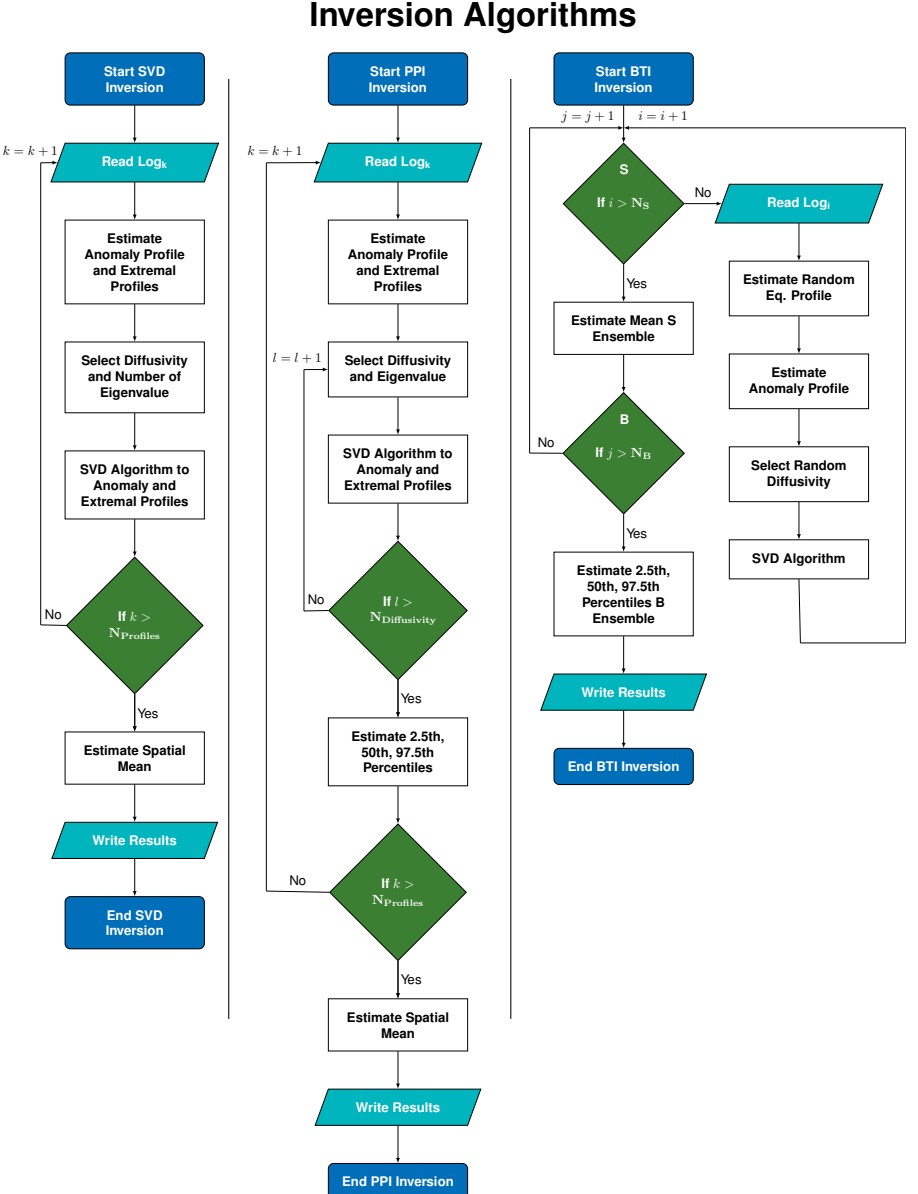

**Figure 2.** Flowchart diagram of the three methods used here to retrieve STP inversions: SVD (left), PPI (center), and BTI (right). All three methods are based on SVD inversions, but considering different approaches to estimate uncertainties. SVD and PPI methods invert three anomaly profiles from each log, the anomaly estimated from the quasi-equilibrium temperature profile and two extremal anomaly profiles estimated using the error in the determination of the quasi-equilibrium temperature profile, which provide the best estimate and the uncertainty of the results (see Figure 1). The PPI technique also considers a range of possible thermal properties and a range of eigenvalues to retrieve the inversions. The BTI method considers a range of possible thermal properties and different quasi-equilibrium temperature profiles, but it aggregates the solutions from considering different values of the parameters (and several logs if applicable) by following a bootstrap sampling strategy, while the SVD and PPI techniques just average the inversions of the three anomaly profiles. See Methods section for more details.





**Figure 3.** Ground surface temperature histories from the artificial profile using different step lengths for retrieving the surface signal (rows), and retaining different number of eigenvalues (columns). Colours indicate the method employed for the estimates: SVD (purple), PPI (red), and BTI (light blue). BTI results are derived using a Sampling ensemble of one member and a Bootstrapping ensemble of 1,000 members. Shadows indicate the uncertainty range for each method. The original surface signal is represented in black.



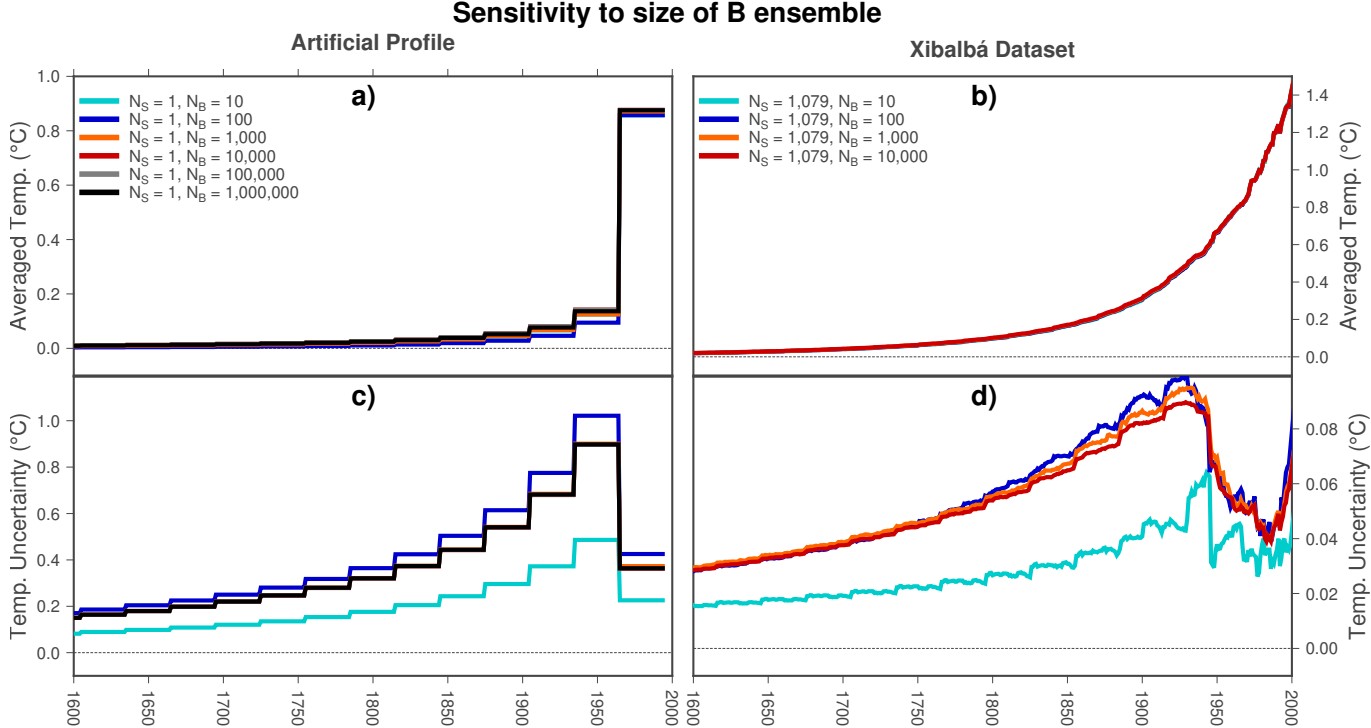

**Figure 4.** Temperature histories and uncertainty ranges in bootstrap inversions for different number of realizations. (a) Temperature histories for inversions of the artificial profile considering a population of one log ($N_S = 1$) and different number of realizations ($N_B$ from 10 to 1,000,000). (b) Temperature histories for inversions of the Xibalbá dataset considering a population of 1,079 logs ($N_S = 1,079$) and different number of realizations ($N_B$ from 10 to 10,000). (c) Uncertainty range for inversions of the artificial profile considering the same sizes for the Bootstrapping ensemble as in panel a. (d) Uncertainty range for inversions of the Xibalbá dataset considering the same sizes for the Bootstrapping ensemble as in panel b.



Figure 5. Global averaged ground surface temperature histories from Xibalbá profiles using different step lengths for retrieving the surface signal (rows), and retaining different number of eigenvalues (columns). Colours indicate the method employed for the estimates: SVD (purple), PPI (red), and BTI (light blue). BTI results are derived using a Sampling ensemble of 1,079 members and a Bootstrapping ensemble of 1,000 members. Shadows indicate the uncertainty range for each method. Note that the uncertainty in BTI results is represented, although difficult to visualize due to its small size.







Figure 6. Global averaged ground heat flux histories from Xibalbá profiles using different step lengths for retrieving the surface signal (rows), and retaining different number of eigenvalues (columns). Colours indicate the method employed for the estimates: SVD (purple), PPI (red), and BTI (light blue). BTI results are derived using a Sampling ensemble of 1,079 members and a Bootstrapping ensemble of 1,000 members. Shadows indicate the uncertainty range for each method. Note that the uncertainty in BTI results is represented, although difficult to visualize due to its small size.



**Figure 7.** (a) Ground surface temperature histories from the three techniques analysed here using the Xibalbá dataset in comparison with previous results from Cuesta-Valero et al. (2021c) (black) and the PAGES2k-Land temperatures (orange). (b) Ground heat flux histories from the three techniques analysed here using the Xibalbá dataset in comparison with results from Cuesta-Valero et al. (2021c) (black). Results from the BTI technique in both panels consider Sampling and Bootstrapping ensembles with 1,079 and 1,000 members, respectively. Note that the uncertainty in BTI results is represented, although difficult to visualize due to its small size.





**Figure 8.** Uncertainty ranges for the three techniques analysed here, using error propagation to estimate the uncertainties in the SVD and PPI global averages. (a) Uncertainty ranges of the global averaged temperature histories from Xibalbá subsurface temperature profiles using error propagation of SVD and PPI inversions (purple and red lines), and standard BTI inversions (light blue lines). (b) Uncertainty range of the global averaged heat flux histories from Xibalbá subsurface temperature profiles using error propagation of SVD and PPI inversions (purple and red lines), and standard BTI inversions (light blue lines). Dashed red and light blue lines in panel a indicate results with constant thermal diffusivity ($\kappa$) for PPI and BTI inversions, respectively. Dashed red and light blue lines in panel b indicate results with constant thermal diffusivity ($\kappa$) and constant thermal conductivity ($\lambda$) for PPI and BTI inversions, respectively. Heat flux estimates using the PPI method consider a constant thermal conductivity, while BTI estimates consider a range of possible thermal conductivities (Table 1).