# Peer review of "A new bootstrap technique to quantify uncertainty in estimates of ground surface temperature and ground heat flux histories from geothermal data"

_Geoscientific Model Development, 2022_

## Author Comment (AC2)

**Dear reviewer:**

**Thank you for your thoughtful and constructive feedback. Here we provide a complete documentation of the changes made in the manuscript in response to each of your comments. Reviewers' comments are shown in plain text, while author responses are shown in bold green text.**

Summary: Past global surface temperatures over the past few centuries can be estimated from present borehole temperature profiles applying inversion techniques based on the solution of the heat transfer equation. Form a set of sites where temperature profiles have been measured, large-scale temperature reconstructions can be derived by averaging the local retrievals. The manuscript presents a method to estimate the uncertainties in the large-scale average estimations based on a bootstrap approach. the authors conclude that this new method provides better and more realistic uncertainty estimates that previous methods. Those previous methods simply calculated the average of the high and low ends of the local uncertainty ranges.

Recommendation: I think that in general the manuscript is valuable and should be published after some revisions. However, I am afraid that one of the motivations of the present study, namely that the previous estimations of uncertainness was unrealistic, contains a conceptual misconception, although it has been previously published. Therefore, the motivation of the present manuscript should be amended to present a correct statistical case. I explain below in more detail my main concern.

1) The manuscript presents a base method to estimate global or large-scale uncertainties that has been published previously. This method just constructs the high-end (and low-end) uncertain range of the global average by calculating the average of the high-end (or low-end) range of the local estimations. This is, however, not correct, as it can be illustrated in a short counter-example. The interpretation of a 5-95% uncertain range in a frequentist approach is that the range covers the true value with 90% probability (technically, it means that a putative infinite number of realizations of the measurements and their corresponding uncertain estimations will contain the true value 90% of the time). For the sake of this reasoning, we can a bit sloppily say that that the probability that the true value is within the estimated uncertainty range is 90%. However, if the uncertainty ranges are constructed by averaging the 5% and the 95% local ranges, this probability is much much larger than 90%. Let us focus on the high end (95%). The probability for the average to be outside that 95% range is not 0.05, but actually 0.05 to the Nth power , where N is the number of profiles (sites).This results because each profile from which that average is constructed, has a probability of 0.05. If N=100, this number is very small, much smaller than 0.05.

The authors realize in the discussion that indeed this estimation is not correct. There, they apply a much

more correct estimation assuming that the global profile is the average of N random variables, and therefore, assuming that these N random variables are independent, the error in the average amounts to the sqrt of the average squared error. If all individual errors are equal, this amounts to that individual error divided by the sqrt(N).

There is one important underlying assumption: the errors should be independent across space. But even if this assumption is not completely fulfilled, this estimation is much more realistic that simply the average of the upper and lower local percentiles, which is clearly incorrect.

Thus, to some extent, the manuscript corrects a previous statistical misconception. In this sense, it is useful, but the motivation of the manuscript should be cast differently,as the reader will be really surprised to see, without any caveat, a clearly wrong method as a benchmark.

**As suggested by the reviewer, we have indicated on the text that the main caveat of the SVD and PPI techniques used in our manuscript is the lack of a correct statistical method to provide with confidence intervals for the average of inversions from several subsurface temperature profiles. We have also changed the aim of the manuscript indicating that we provide a new, revised and improved method to aggregate inversions from different subsurface temperature profiles.**

On the other hand, the question of the spatial correlation of uncertainties, which is critical for the validity of both methods (bootstrap and error propagation) is not mentioned at all.

The bootstrap approach is definitively better - and I could not see any clear error in this application of bootstrapping. However, this approach does not take into account the possible spatial correlation of the local errors. I do not know how significant these correlations might be, but if they are, then the bootstrap estimation of the uncertainty will be too narrow –in the same as the error propagation proposed by the authors in the discussion– since the effective number of degrees of freedom will not be N, but smaller.

If these correlations are relevant, the bootstrap should take it into account, e.g. by block-bootstrap, in which correlated regions are first averaged together, and then bootstrapped. I think that this problem is technically very difficult to solve satisfactorily, but again, I believe that the presented bootstrap approach is indeed useful.

**Indeed, spatial correlation may be important to determine the confidence interval of the global estimates of temperature and heat flux change from subsurface temperature profiles. To account for the effect of spatial correlation on global averages, we have estimated the effective degrees of freedom of surface air temperatures at each grid cell of the CRU TS 4.05 product (Harris et al., 2020).**

The degrees of freedom (dof) of two temporal series depends on the correlation coefficient ($c$) between them (Fraedrich et al., 1995) as

$$\mathrm{dof} = \frac{2}{1 + c^2}. \tag{1}$$

In order to estimate the spatial variation of the degrees of freedom of CRU temperatures, we apply Equation (1) to the temperature series in a given cell and the four closest neighbours, obtaining the effective degrees of freedom as the average of the four different estimates. Figure 1 in this document shows the spatial degrees of freedom for annual temperature series and 30-yr running means generated by repeating this process for all grid cells in the CRU product. Results considering the eight and twelve closest neighbours are also displayed. Orography seems to be the leading factor in local variability, with the small number of observations included in the product for several areas, like the Arctic and Africa, also displaying an effect.

We include the different effective degrees of freedom in the bootstrap estimates by estimating the weighted mean of the inversions in the Sampling ensemble to retrieve the corresponding member of the Bootstrapping ensemble. That is, the inversions within the Sampling ensemble are weighted by the corresponding degrees of freedom at the location of the profile, thus inversions from temperature profiles within zones with high degrees of freedom weight more than inversions from profiles in other zones. Concretely, we consider the degrees of freedom obtained using the twelve closest neighbours and 30-yr running means, as this is the case showing higher zonal differences in Figure 1. However, the retrieved global averages and 95% confidence intervals from bootstrap inversions including the different effective degrees of freedom and without considering them present very similar results (Figure 2). Additionally, similar results are obtained when considering annual temperatures, and four and eight neighbours (not shown). Therefore, the effect of the different degrees of freedom at borehole locations is not particularly large for global temperatures retrieved from temperature profiles. We have added a Supplementary information document to the manuscript including these points, as well as Figures 1 and 2 in this document.

Particular points:

2) A definition of the quasi-equilibrium temperature will help some readers.

We have added few lines describing the quasi-equilibrium temperature profile and its main characteristics on the new version of the manuscript.

**Effective degrees of freedom**

[Figure]

**Figure 1:** Effective degrees of freedom for CRU TS 4.05 temperatures from annual (left column) and long-term (30-yr running means, right column) series. Results considering the four (first row), eight (second row), and twelve (third row) closest grid cells are also displayed.

**Effect of E.D.O.F.**

**Figure 2:** Estimated temperature evolution from subsurface temperature profiles. (a) Global averaged surface temperature histories considering the different effective degrees of freedom at the location of each profile (red line), and weighting all profiles equally (purple line). (b) Range of the 95% confidence interval for bootstrap inversions considering the effective degrees of freedom at the location of each profile (red line), and weighting all profiles equally (purple line).

3) Section 3.5, perhaps the most important section, is not very clearly written (I needed to read it several times). For instance , line 257: "named Sampling and Bootstrapping ensembles (S and B ensembles in Figure 2). The Sampling ensemble consists ...". and the reader expect the following sentence to explain what the Bootstrapping ensemble is. However, the text goes on with "The BTI method considers the uncertainty arising from ...". This is quite confusing. Actually, the bootstrapping ensemble is a typical bootstrap sampling from the set of individual local profiles, where each profile has been derived from one value of the uncertain parameters (T0, Gamma0, and thermal conductivity). The only restriction is that each sites contributes with one member to the ensemble.

All in all, I found the technical description unnecessarily too cumbersome.

**We have modified Section 3.5 in order to improve the clarity of the text. Please, see this section on the new version of the paper.**

**References**

Fraedrich, K., Ziehmann, C., and Sielmann, F. (1995). Estimates of Spatial Degrees of Freedom. *Journal of Climate*, **8**(2), 361 –369. DOI: 10.1175/1520-0442(1995)008<0361:EOSDOF>2.0.CO;2.

Harris, I., Osborn, T. J., Jones, P., and Lister, D. (2020). Version 4 of the CRU TS monthly high-resolution gridded multivariate climate dataset. *Scientific Data*, **7**(1), 109. DOI: 10.1038/s41597-020-0453-3.

---

## Author Comment (AC3)

**Dear reviewer:**

**Thank you for your thoughtful and constructive feedback. Here we provide a complete documentation of the changes made in the manuscript in response to each of your comments. Reviewers' comments are shown in plain text, while author responses are shown in bold green text.**

In this article, the authors introduce a new technique to estimate ground surface temperature and ground heat flux histories. Using an artificial temperature profile and real data from the Xibalbá subsurface temperature profiles, they show how this new technique performs against the tested singular value decomposition and perturbed parameter inversions methods. Little difference is noted when examining individual profiles. However, notable differences in the uncertainty range for the inversion of large sets of subsurface temperature profiles are observed with the new bootstrap technique producing smaller uncertainty ranges.

The authors clearly explain the importance of these various methods and shedding light on the thermal state of the land surface. They elaborate the three different methodologies and provide useful figures to illustrate their points. I found Figure 2 to be especially helpful. I believe that the outlined new methodology is a useful new tool for the community and would recommend publication after some minor changes.

Comments:

There is little discussion in the data section as to the uncertainty associated with the proxy-based temperatures and the Xibalbá subsurface temperature profiles. Are there errors associated with converting the PAGES2k global temperatures to a land temperatures? With respect to the Xibalbá subsurface temperature profiles, they haven't all been measured at the same time. This will influence the reconstruction. I think a couple of lines here elaborating these uncertainties could be helpful. It would also help the reader further understand the complexity of undertaking these inversions as the data is not perfect and that any tool that can minimize uncertainty is important.

**We have explained how the uncertainty of the PAGES2k temperature series is transformed for the PAGES2k-Land temperatures, and we have expanded the description of the Xibalbá profiles in the new version of the manuscript including the points raised by the reviewer.**

On L153 and later in Section 3.8, the authors state how the continental subsurface is considered homogeneous (i.e. thermal properties are constant) and give the example of the Arctic. However, I cannot believe that the Arctic could be considered representative of the globe. How can one consider a constant thermal diffusivity and conductivity for all regions? Is this an unfortunate trade off due to the lack of

subsurface thermal data?

**Indeed, only a few temperature profiles provide information about the thermal properties of the subsurface. However, the logs including information about thermal properties, like the example cited in the manuscript, show variations with depth around a mean value, thus the assumption of an homogeneous medium is reasonable. Furthermore, changes in subsurface thermal properties incompatible with an approximately homogeneous subsurface would appear in the recorded temperatures as a non-climatic signal, and such log would not be used in the analysis as explained in Section 2.2 of the manuscript. It is also worth to mention that the Xibalbá database excludes temperature records above $20\,\mathrm{m}$ of depth, thus we only consider temperature measurements in bedrock. Therefore, the factors accounting for the largest variations in thermal properties, such as soil texture and composition, as well as soil moisture changes, are not relevant for the inversions. For these reasons, we can safely consider the subsurface as an homogeneous medium. We have tried to clarify this in the new version of the manuscript.**

On L418, the authors state: "Another remarkable results is the agreement between the PAGES2k land temperatures and the ground surface temperature histories for most of the period." While this appears to be true for a quick glance at Figure 7, notable differences can be seen when examining Table 2. The three methods reconstruct a warming about two times greater in 1950-2000, 1900-1950, 1850-1900, and 1800-1850. The authors do note this following L418. However, I believe this sentence should be rephrased to emphasize the how the tendency is captured but not the magnitude and/or highlight the excellent job the methods do in reconstructing the period of 1600-1800.

**We agree that speaking in terms of trends is more accurate in this context, we have modified the text accordingly.**

In Figure 8b, PPI and BTI with varying $\kappa$ and $\lambda$ show an increase in heat flux as of about 1970 that is not observed in SVD nor PPI and BTI with constant $\kappa$ and $\lambda$. This is not elaborated in the text. It would be helpful to a reader to have a couple lines clarifying this.

**Figure 8 does not show surface temperature histories nor heat flux histories, but the uncertainty ranges in temperature and heat flux histories. In any case, the reviewer is right, the effect of varying thermal properties in Figure 8 is not discussed on the text. In fact, this effect does not affect the conclusions drawn from the results displayed in this figure, and therefore we have removed the lines that are not strictly required from the figure.**

Technical Points:

L51-52: Should read: "Nevertheless, several sources of uncertainty arise in the inversion process, the most important being..."

Overall, watch the use of "consists in" in the text. For the majority of time used, it should actually be "consists of"

L59-60: Should read: "...the deepest part of the observed profile, then providing a best estimate..."

L101: "These experiments also allow to identify..." should read "These experiments allow for the identification of..."

L153: Should read: "...typical vary by a relatively small..."

L196: Should read: "Finally, small eigenvalues are from S-1 in order to stabilize the solution..."

L204-205: Should read: "The errors in the estimates of the long-term surface temperature (T0) and the geothermal gradient ($\gamma$0) have..."

L251: Should read: "As explained in the Introduction, the SVD and PPI techniques do not provide a comprehensive..."

L263: "...which are retrieved form..." should be "which are retrieved from..."

The units of thermal diffusivity are shown as m2s but they should be m2s-1

L394: sensible should be sensitive

L433: "This three factors..." should be "These three factors..."

**We thank the reviewer for taking the time to annotate all these issues. We have reviewed the text and have addressed all points.**

In Table 1, please define Ns and NB

**We have defined these parameters in the new version of the manuscript.**

In Figure 3, the shading associated with the purple line isn't purple but blue. For consistency with the other figures, I recommend using the same purple shading colour as the other figures.

**The color of the SVD shading looks different in Figure 3 than, for example, in Figure 5. Nevertheless, this is not because different colors are used for the SVD shading in these figures, but**

because the order in which the shadings are drawn is different, affecting the displayed color. For example, Figure 6f shows that the shading of the SVD line is purple when drawn alone, but if drawn before the PPI shading, it looks as a markedly different shade of purple. Please, note that we change the drawing order to facilitate the interpretation of the figures, since the final color scheme if we follow the same drawing order in Figures 3 and 5, for example, would not provide the same level of detail. We could also change the colors used in the figures, but this will not solve the problem with the drawing order, and we would be using different colors to represent the same thing across the figures. We tested different colors and drawing orders, but we think it is best to keep the same colors though the entire manuscript, although the final colors of the shadows in Figure 3 seem different.

In the title of Figure 4, I would clarify that B in the title stands for Bootstrapping.

We have changed the "B" in the figure title by "bootstrapping" in the new version of the figure.

---

## Author Response (AR2)

Dear GMD editorial team:

We thank the reviewers for taking the efford to read our manuscript a second time.

This *Response to reviewers* file provides a complete documentation of the changes made in response to reviewer 1's comments. Reviewers' comments are shown in plain text. Author responses are shown in bold green text. Corrections within the revised manuscript are shown in blue text. All line numbers in this file refer to locations in the revised manuscript with changes marked unless indicated otherwise.

**Reviewer 1**

The methodology outlined in the article is a useful tool for the community and I believe that the authors have sufficiently addressed the points brought up during the revision process. I recommend that it is accepted after some minor technical changes (noted below).

Technical Points:

L14: "Furthermore, the new bootstrap technique provides with a..." should read "Furthermore, the new bootstrap techniques provides a..." or "Furthermore, the new bootstrap techniques provides us with..."

L31: "...,which allow to estimate..." should read "...which allow for the estimation..." or "...which allow us to estimate..."

L177: "...time series Equation (3)..." I believe there should be a period here. "...time series. Equation (3)..."

L407: "...which provides with a conservative..." should read "...which provide a conservative..." or "...which provide us with a conservative..."

L421: "...does not change much the results..." should read "...does not change the results much..."

L474: "...to provide with..." should read "...to provide..."

L478: "...PPI uncertainties are difficult to interpreted..." It should be interpret.

L515: "…although providing with…" The with should be removed.

**We thank the reviewer for noting this points. We have addressed all of them in the new version of the manuscript.**